# A Summer of Cyanobacterial Blooms in Belgian Waterbodies: Microcystin Quantification and Molecular Characterizations

**DOI:** 10.3390/toxins14010061

**Published:** 2022-01-16

**Authors:** Wannes Hugo R. Van Hassel, Mirjana Andjelkovic, Benoit Durieu, Viviana Almanza Marroquin, Julien Masquelier, Bart Huybrechts, Annick Wilmotte

**Affiliations:** 1Toxins Unit, Organic Contaminants and Additives, Sciensano, Rue Juliette Wytsmanstraat 14, 1050 Brussels, Belgium; Julien.Masquelier@sciensano.be (J.M.); Bart.Huybrechts@sciensano.be (B.H.); 2InBios-Centre for Protein Engineering, Department of Life Sciences, University of Liège, Allée du Six Août 11, 4000 Liège, Belgium; benoit.durieu@uliege.be (B.D.); awilmotte@uliege.be (A.W.); 3Risk and-Health Impact Assessment, Sciensano, Rue Juliette Wytsmanstraat 14, 1050 Brussels, Belgium; Mirjana.Andjelkovic@sciensano.be; 4Phytoplankton and Phytobenthos Laboratory, EULA Center, University of Concepcion, Barrio Universitario Box 160, Concepcion 3349001, Chile; valmanza@udec.cl

**Keywords:** planktonic cyanobacteria, microcystin, blooms, monitoring, analysis, mass spectrometry, Liquid Chromatography with tandem mass spectrometry (LC-MS/MS)

## Abstract

In the context of increasing occurrences of toxic cyanobacterial blooms worldwide, their monitoring in Belgium is currently performed by regional environmental agencies (in two of three regions) using different protocols and is restricted to some selected recreational ponds and lakes. Therefore, a global assessment based on the comparison of existing datasets is not possible. For this study, 79 water samples from a monitoring of five lakes in Wallonia and occasional blooms in Flanders and Brussels, including a canal, were analyzed. A Liquid Chromatography with tandem mass spectrometry (LC-MS/MS) method allowed to detect and quantify eight microcystin congeners. The *mcyE* gene was detected using PCR, while dominant cyanobacterial species were identified using 16S RNA amplification and direct sequencing. The cyanobacterial diversity for two water samples was characterized with amplicon sequencing. Microcystins were detected above limit of quantification (LOQ) in 68 water samples, and the World Health Organization (WHO) recommended guideline value for microcystins in recreational water (24 µg L^−1^) was surpassed in 18 samples. The microcystin concentrations ranged from 0.11 µg L^−1^ to 2798.81 µg L^−1^ total microcystin. For 45 samples, the dominance of the genera *Microcystis* sp., *Dolichospermum* sp., *Aphanizomenon* sp., *Cyanobium/Synechococcus* sp., *Planktothrix* sp., *Romeria* sp., *Cyanodictyon* sp., and *Phormidium* sp. was shown. Moreover, the *mcyE* gene was detected in 75.71% of all the water samples.

## 1. Introduction

A Belgian global picture of the diversity of cyanotoxins and the taxa producing them is currently lacking. This impairs a better understanding of their importance, their yearly variations and the need to prevent and mitigate blooms. Drinking water in Belgium is mostly provided by aquifers and barely depends on reservoirs [1,2]. Moreover, the water supply varies among the regions (Brussels regions, Flanders, and Wallonia). In Flanders, besides aquifers, the Meuse river and the Albert canal, only eight reservoirs supply water. If this is not sufficient, water is imported from Wallonia and neighboring countries [2]. The drinking water in Brussels originates mainly (>90%) from Wallonia [3]. Water exploitation in Wallonia is dependent on ground water for 80% and is supplemented with water exploitation from the Meuse, some old mining sites and six dams [1,4,5]. Further, recreational waters are a sensitive issue, as there is an increasing societal demand for such areas in summer. External sources of eutrophication, such as untreated sewage discharge or agriculture run off, in these waterbodies may promote cyanobacterial bloom formation [6,7]. However, little information about eutrophication sources is available for fresh waterbodies in Belgium.

Ingestion of cyanotoxin contaminated water has been shown to be detrimental to human and animal health [8,9,10,11,12,13,14]. Yet in Belgium, no causative link has so far been found between toxic blooms and associated symptoms in humans and animals, such as gastroenteritis, vomiting, liver damage or convulsions [15,16]. However, suspicious bird deaths have been reported, which coincided with a toxic *Microcystis* bloom [17].

Between 1994 and 2008, a few studies identified the morphological and toxin diversity in toxic cyanobacterial blooms in Belgian lakes and ponds [17,18,19,20,21,22,23,24]. The concentrations of total microcystin (MC) measured by Ultra High Performance Liquid Chromatography (UHPLC) reached 18 to 2651 µg g^−1^ dry weight (DW), and the bloom samples contained up to six variants [20,22]. *Microcystis* was found to be the most dominant genus, followed by *Planktothrix*. These results prompted a large scale study (BelSPO project B-BLOOMS2) from 2007 to 2010 [23,24]. During this study, 89% and 83% of the 162 samples tested showed the presence of *mcyE* and *mcyA* genes, respectively. *Microcystis*, *Anabaena* (now taxonomically identified as *Dolichospermum*), *Aphanizomenon*, *Planktothrix, and Woronichinia* were the primary bloom-forming cyanobacterial taxa. Furthermore, a quantitative toxin analysis of the samples showed that the total congeners concentration varied from 0.120 µg L^−1^ to 37500 µg L^−1^ total microcystin, analyzed with high performance liquid chromatography—photodiode array detection (HPLC-DAD) and ELISA, in parallel [23,24,25]. Based largely on the B-BLOOMS2 report, the public authorities started to take action by informing the citizens and including cyanobacterial blooms in monitoring studies. Presently, Flemish and Walloon environmental agencies perform limited monitoring of recreational ponds (where bathing or other activities involving water contact are allowed) but using different protocols. This approach precludes the possibility of obtaining a global overview of blooms in Belgium. Furthermore, the data is limited to a small number of waterbodies, as bathing in Belgian surface waters is very restricted. Our study will extend the data from the B-BLOOMS2 study by providing new toxin, molecular and cyanobacterial occurrence data for water samples after a 10-year hiatus, using a uniform protocol. Moreover, the MC quantification revealed the existence of new microcystin congeners.

MCs belong to the most common cyanotoxins group found worldwide and are produced by multiple taxa (e.g., *Microcystis aeruginosa*, *Planktothrix* sp., *Anabaena*/*Dolichospermum*, *Oscillatoria* and *Nostoc*) [26,27]. The MCs covalently bind the protein phosphatases 1 and 2A (PP1 and PP2A) in eukaryotes, inhibiting their functions and eventually causing cell death [28,29,30]. Upon ingestion by mammals, the congeners are primarily transported in the liver cells through specific organic anion transporting polypeptides (OATPS) [31,32,33,34], which results in a hepatotoxic effect, causing nausea, intestinal problems and liver damage [29,30,35,36,37,38,39,40,41]. These toxins can also effect other organs such as the lungs and kidneys [34,39,42]. Human exposure to the MCs through multiple routes have been described (e.g., drinking water, recreational exposure, cyanobacteria-based food supplements, contaminated crops, …) [14,16,39,41,43,44,45,46,47,48,49]. The most prevalent congener found in Europe is MC-LR, though it is rarely detected in isolation [21,27,50,51,52,53,54,55]. Therefore, MC-LR is also commonly used in toxicological assays [36,37,40,56,57]. However, other congeners have single or multiple modifications in their structure, such as different amino acids in positions two and/or four, or methylations at different positions [58,59,60]. The structurally different congeners interact differently with the OATPS, PP1 and PP2A, resulting in different toxicities [31,58,61,62,63,64]. The half maximal inhibitory concentration (IC_50_) for PP2a and lethal dose for half of the test population (LD_50_) in vivo for MC-RR are shown to be lower than MC-LR [61,62,64]. More efficient uptake for MC-LF and MC-WR than MC-LR is suggested to correspond to higher in vivo toxicity, while the PP-inhibiting capabilities are comparable [31,34,61,62]. Although differences in toxicity between the congeners are known, no uniform toxicity equivalency factors are available to adjust for the variation in their activity, as is the case for marine toxins [65,66]. An accurate risk assessment, when several congeners are present, is, therefore, difficult. Thus, the World Health Organization (WHO), Environmental Protection Agency (EPA) and other regulatory agencies use the sum of the concentrations for all MCs, described as MC-LR equivalent (MC-LR Equiv). The WHO published, in 1994, an initial tolerable daily intake (TDI) guideline of 0.04 µg kgbodyweight^−1^ day^−1^, which translated to a concentration of 20 µg L^−1^ in surface waters used for recreational activities [67]. In 2020, the WHO updated its provisional guideline value to 24 µg L^−1^ [39]. The changed value results from a difference in calculations. The original value (20 µg L^−1^) was based on the proposed TDI, the body weight of an adult and the involuntary ingestion of 100 mL of water during swimming activities [68]. The new guideline value is calculated from the no observed adverse effect level (NOAEL) (40 µg kg^−1^ bodyweight) [57], a ten times reduced uncertainty factor compared to the proposed TDI due to the short-term nature of recreational exposure, a volume of 250 mL of involuntarily ingested water, and taking into account the bodyweight of a child instead of an adult. The WHO guideline values are calculated to provide an adequate margin of safety [39]. The US E.P.A. also provided a new guideline for recreational waters in 2019. A reference dose (RfD) of 0.05 µg kg bodyweight^−1^ day^−1^, the mean body weight of children between 6 and 10 years and an incidental ingestion factor were used to calculate 8 µg L^−1^ as the recommended value [69].

Since the B-BLOOMS2 study was finalized a decade ago, no studies with standardized protocols have been performed to monitor cyanobacterial diversity and toxins in water samples from all over Belgium. For the first time in Belgium, we utilized Ultra High Performance Liquid Chromatography with tandem mass spectrometry (UHPLC-MS/MS) to identify and quantify the most frequent microcystin congeners in the water samples. By also detecting the genetic potential for synthesizing MCs and the dominant species in the samples, we tried to obtain more insights concerning the bloom characteristics in Belgium. Our cooperation with the three regional environmental agencies in Belgium achieved a sampling on a wide spatial scale of 79 water samples, covering 23 aquatic ecosystems. The species and toxin profiling by a standard set of analyses reveal the importance of monitoring in space and time. Furthermore, by targeting various waterbodies, we aim to identify the locations where monitoring would be needed because there is a health risk. Additionally, the data could help to design more effective prevention and mitigation measures.

## 2. Results

### 2.1. Toxin Quantification

In 86.08% (68/79) of the water samples, at least one of the quantified MCs (MC-RR, MC-LA, MC-LF, MC-LR, MC-LY, MC-LW, MC-YR, MC-WR) could be found above the limit of quantification (LOQ = 12.5 µg kg^−1^ before correction for the total sample weight and volume). Moreover, 22.78% (18/79) of the samples contained toxin concentrations higher than 24 µg L^−1^ total microcystin [39]. A complete overview of the results can be found in Appendix A. The concentration range was between 0.11 µg L^−1^ and 2798.81 µg L^−1^ total microcystin.

Furthermore, 82.61% (19/23) of the waterbodies contained, at least once during the summer, a quantifiable concentration of toxin (>LOQ), whereas concentrations higher than 24 µg L^−1^ total microcystin were detected in 34.78% (8/23) of the waterbodies.

Interestingly, the canal samples (BV1, BV2, BV3) from Brussels contained congener concentrations higher than LOQ. One of the samples even reached 1831.32 µg L^−1^ total microcystin. These results are the first reports on blooms in Belgium waterways.

Validation results for the UHPLC-MS/MS method used for analysis of the water samples can be found in Appendix A.

### 2.2. Toxin Congener Diversity

The detection frequency of MC-RR was the highest (84.81%), followed by the detection frequencies of MC-LR (81.01%) and MC-YR (50.63%).

MC-LR, MC-RR and MC-YR also contribute the most to the total MCs concentration in individual samples, compared to the other congeners. When comparing the proportions of the highest contributors in Belgium, the proportion of MC-RR is significantly higher than MC-LR, based on the Wilcoxon signed-ranked test (α < 0.05). Proportionally, MC-LR is the second-highest contributor, followed by MC-YR. The Wilcoxon test shows a significant difference (α < 0.05) between the proportional contributions of MC-YR compared to the other congeners (Figure 1a). Separate statistical analysis of samples containing a total MC concentration above or below the WHO guideline value for recreational water showed that the proportions of MC-YR are also significantly lower in relation to the proportions of MC-LR and MC-RR both above or below the 24 µg L^−1^ total microcystin (Figure 1b,c).

When comparing samples with a total MC concentration above or below the guideline value, several observations can be made. There was a significant difference in MC-YR (Figure 1f) but no significant difference in the proportions of MC-LR and MC-RR (Figure 1d,e). Comparing the two concentration ranges for the proportions of MC-LA, MC-LY, MC-LF and MC-LW, a significant difference was shown using the Wilcoxon signed-rank test (α < 0.05) (data not shown). A higher diversity of congeners contributed to the total MCs concentration when the concentration was above the WHO guideline value for recreational water (24 µg L^−1^ total microcystin) (Figure 1b,c).

Additionally, the water samples were screened for six other MCs (MC-HtyR, dm-MC-LR, D-asp- MC-LR, dm-MC-RR, D-asp-Dhb-MC-RR and MC-HilR), which are also commonly detected in other studies [20,70,71]. These toxins were not included in the initially designed validation process. However, due to their prevalence and possible toxicity, they were screened. The congeners are identified based on molecular mass, production ions and elution time with the UHPLC-MS/MS method. However, dm-MC-LR and D-asp- MC-LR as well as dm-MC-RR and D-asp-Dhb-MC-RR could not be separated based on these parameters and are reported together (Table 1). We further establish limits of detection for the congeners, shown in Table 1. Overall, dm-MC-RR/D-asp-Dhb-MC-RR were the most abundant congeners in the water samples, followed by dm-MC-LR/D-asp- MC-LR, MC-HilR and MC-HytR, sequentially. An overview of their detection frequency can also be seen in Table 1, as well as their frequency related to the total quantified microcystin concentration in the samples. A complete overview of the results per sample can be found in Appendix A.

### 2.3. Molecular Analysis of Water Samples

PCR amplification of the 16S rRNA fragment was attempted for 76 water samples. The fragment was successfully amplified for 45 samples. Some water samples were amplified twice (e.g., BL1.29, BV1.34, B04.29, I04.32), as can be seen in Appendix A. Direct Sanger sequencing of the 16S rRNA fragment from the water samples resulted in 49 sequences of sufficient quality that could be analyzed with BLAST. They were of cyanobacterial origin, except for one plastid sequence in sample E04.32. The majority (42/49) of the analyzed 16S rRNA fragments showed 97% or higher similarities to sequences found in Genbank, as shown in Appendix A. However, not all samples had a single dominant species, and 31 failed sequencings corresponded to mixtures of sequences that could not be analyzed. In four cases, the PCR with different primer pairs gave different dominant genera, and both are shown in Appendix A. For all the samples analyzed with direct Sanger sequencing, *Microcystis* was the most dominant genus (12/76 or 15.79%), closely followed by *Dolichospermum* (11/76 or 14.47%). The third most abundant genus was *Aphanizomenon* (8/76 or 10.53%). Furthermore, the *Synechococcus* and *Planktothrix* genera were dominant in five and three samples, respectively. The *Cyanobium* genus was observed twice, while the *Cyanodictyon, Romeria* and, *Phormidium* genera were found once.

The amplicon sequencings by Illumina indicated a dominance of sequences from five OTUs belonging to the *Dolichospermum* genus for sample BL5.29 (71% of the reads), followed by *Microcystis* (20.5% of the reads) and a minor fraction of *Aphanizomenon* and *Cyanobium*/*Synechococcus*. This corresponds to the dominance of *Dolichospermum* inferred from the direct sanger sequencing. In contrast, sample VL1.36 was completely dominated by sequences of *Microcystis* (99.1% of the reads) followed by 0.6% of the reads affiliated to *Dolichospermum*. However, the direct Sanger sequencing did not give any readable sequence to compare (Appendix A).

The dominance of *Microcystis* in VL1.36 coincides with a high diversity (seven microcystin congeners) and a high concentration of total microcystin (128.93 µg L^−1^). In contrast, only two congeners and a lower total microcystin concentration (1.22 µg L^−1^) were found in the *Dolichospermum* dominated bloom BL5.29 (Appendix A).

The *mcyE* gene amplification was tested in 70 water samples. The *mcyE* gene was detected together with MCs in 71.43% of the samples. Moreover, 4.29% of the samples contained the *mcyE* gene though the presence of MCs could not be detected by UHPLC-MS/MS method. However, the presence of the *mcyE* gene only implies the potential to produce MCs. Microcystin will not be produced when *mcyE* or other genes of the *mcy* gene cluster are lacking, are silenced or contain mutations.

### 2.4. Cyanobacteria Dominance at Different MCs Concentrations

In the water samples, there was a dominance of the genera *Microcystis*, *Dolichospermum*, *Cyanobium/Synechococcus*, *Aphanizomenon* and *Planktothrix*. Sequences affiliated with *Cyanodictyon*, *Romeria*, and *Phormidium* were each observed in one of the samples. In samples with quantified MCs below the WHO guideline value, *Aphanizomenon*, *Dolichospermum*, *Microcystis*, *Synechococcus*/*Cyanobium*, *Planktothrix* and a plastid were identified as dominant (Figure 2a). For water samples that contained MCs concentrations above the WHO guideline value for recreational waters, primarily *Microcystis* or *Dolichospermum* could be identified as dominant species, based on the direct Sanger sequencing (Figure 2b). However, in both cases, it was not always possible to determine the dominant taxon, as the direct sequencing was not successful. When MCs were present in samples where *Dolichospermum* was dominant, concentrations ranged between 0.67 to 2420.91 µg L^−1^ total microcystin, while for samples with *Microcystis* as dominant species, concentrations ranged from 1.07 to 2798.81 µg L^−1^ total microcystin. Samples, dominated by *Aphanizomemon*, contained concentrations between 0.11 and 4.35 µg L^−1^ total microcystin. In two water samples from lake H02, where *Planktotrix* was dominant, total microcystin concentrations were 4.05 and 2.80 µg L^−1^. In contrast, a concentration of 250.35 µg L^−1^ total microcystin was quantified in a water sample from lake I04 containing *Planktothrix*. However, the origin of the MCs can be debated as a week earlier, *Microcystis* was abundant in lake I04, and the bloom could have been in decline, as further shown in Figure 3a. Our results cannot support definitive conclusions that would link high toxin concentrations to a specific cyanobacterial taxon. However, higher total microcystin concentrations are observed when *Microcystis* or *Dolichospermum* are the dominant taxa.

### 2.5. Monitoring of Walloon Recreational Lakes

The weekly sampling of the Walloon lakes provided an opportunity to look at the evolution of the toxin concentration and dominant species with time. The samples from lake I01 (Falemprise) showed a total microcystin concentration higher than 1 µg L^−1^ (I01.31) only once, and when the direct Sanger sequencing was possible, the dominant cyanobacteria belonged to the unicellular *Synechococcus*/*Cyanobium* (Appendix A). This lake was designated as a reference recreational lake during the B-BLOOMS2 study and has been regularly monitored since then. In lake B04 (Renipont plage), the concentration of MCs only rose slightly above 1 µg L^−1^ in three instances when the potentially toxic cyanobacteria genera, *Aphanizomenon* and *Planktothrix* were found (Appendix A). MC concentrations in the samples from H02 (Saint Léger sport complex) never reached the WHO guideline for recreational use but were slightly increasing over the summer and peaked at 4.35 µg L^−1^ at the end of August, coinciding with the presence of the potentially toxic *Aphanizomemon* genus. However, the values decreased in the following weeks (Appendix A). The two lakes where the WHO guideline value was exceeded were E04 (Grand Large, Mons) and I04 (Lac de Bambois). There was an increase in MCs over the summer, ending with a decrease in September in these samples. However, in the latter lake, the MC values were much higher, reaching 250.35 µg L^−1^, and the decrease was more gradual. For both lakes, *Microcystis* sp. was prevalent in the samples just before the MC increases (Figure 3a,b). However, in lake I04, a higher diversity of potentially toxic genera was detected by direct Sanger sequencing, *Aphanizomenon*, *Dolichospermum* and *Planktothrix* (Figure 3a).

## 3. Discussion

For the first time since the B-BLOOOMS2 study, microcystin congeners in Belgian surface waters have been reliably quantified by a standardized, state of the art analytical method in all three Belgian regions. Moreover, unmonitored waterbodies in the Brussels region were included in the study and provided additional proof of toxic blooms.

True monitoring data was achieved in Wallonia due to the weekly sampling of five recreational lakes during the summer of 2019. Three recreational waters (H02, I01, B04) showed MC values lower than the WHO guideline. Falemprise (I01) and Bambois (I04) had already served as a reference recreational lake with regular monitoring and a sporadically analyzed lake, respectively, during the B-BLOOMS2 study. Falemprise had shown quite a variable planktonic diversity, with a regular presence of potentially toxic taxa, the *mcy* genes and total microcystin concentrations ranging from 0.12 to 6.11 µg L^−1^. In Bambois, *Anabaena* (now *Dolichospermum*) formed blooms with MCs concentrations lower than 2.6 µg L^−1^ [23]. In the present study, both lakes contained MCs, showing a persistent problem. Lake Bambois presented the highest values for the five Walloon lakes and exceeded the WHO guideline value for recreational use between the end of July to the middle of September (weeks 31 to 37). These results seem to indicate an increase in toxin concentration during the bloom events. *Aphanizomenon* and *Dolichospermum* were observed before the start of the MCs peaks.

Globally, 34.78% of the studied waterbodies contained total microcystin concentrations higher than the suggested guideline for recreational waters. The highest values were observed in waterbodies in Flanders. These samples also contained the highest fresh weights of filtered biomass (20.4 10^−6^ g L^−1^ average) compared to Brussels (4.6 10^−6^ g L^−1^ average) and Wallonia (2.3 10^−6^ g L^−1^ average). However, high concentrations of total microcystin could not always be linked to a high fresh weight of the biomass, as shown in Appendix A. The total microcystin range (0.11 µg L^−1^ to 2798.81 µg L^−1^) found in 68 of the 78 analyzed water samples is similar to the one found a decade ago during the B-BLOOMS2 study (0.120 µg L^−1^ to 37,500 µg L^−1^ total microcystin) by HPLC or ELISA [23]. The specific screening can explain the difference in certain MCs, resulting in underestimating the actual concentration, as the screening method only includes a selection of congeners and might miss certain ones. The ELISA assays, performed during the B-BLOOMS2 study, have the advantage of detecting all the possible hydrophobic adda groups specific for MCs (Appendix A) in the samples. In contrast, our triple quadrupole MS/MS method is very specific and will detect only the targeted toxins selected a priori in the detection method. This type of method is not well suited for a full scan analysis of samples. Worldwide, total microcystin concentrations can reach a maximum of 42.7 mg L^−1^, although concentrations are usually lower [51,52,53,54,70,72,73,74,75,76,77,78,79,80]. For instance, during a sampling campaign in the United Kingdom, only 18% of the samples contained a concentration above 20 µg L^−1^ total microcystin [51].

In terms of congener diversity, MC-RR (84.81%), MC-LR (81.01%), and MC-YR (50.63%) were most prevalent in our study. Earlier, MC-LR was found to be the dominant congener (in 64% of analyzed water samples) in Wallonia and Luxemburg by Willame et al. [20]. MC-YR was the second most common congener, while MC-RR was not found [20]. Other reports confirm the presence of these most abundant congeners (MC-LR, MC-RR and MC-YR) in western Europe [26,51,53,54,70,71,81,82,83].

Moreover, five MCs (MC-LA, MC-LF, MC-WR, MC-LW and MC-LY) were also quantified in lower concentrations, while six other MCs (MC-HytR, dm-MC-LR, D-asp- MC-LR, dm-MC-RR, D-asp-Dhb-MC-RR and MC-HilR) were only screened and not quantified during our study. For the five quantified MCs, only a minor contribution to the total microcystins concentration at different concentration levels was shown. The MCs that were not quantified were primarily present in samples with total microcystin concentrations above the WHO guideline limit (24 µg L^−1^). The lower contribution or presence of these eleven congeners at higher total microcystin concentrations are consistent in the samples from independent waterbodies and suggest that the biosynthesis dynamics of these toxins is different compared to the more abundant congeners (MC-LR, MC-RR, MC-YR). However, shifts in environmental factors, cyanobacterial strain diversity, growth phase and toxin production dynamics could influence the overall toxin diversity and quota in the blooms [84].

Overall, the minor contribution or presence of these eleven MCs (quantified or not) to the total MC concentration might be of minor importance for the degree of health risk compared to the more abundant MCs (MC-LR, MC-RR and MC-YR). In most cases, the sum of the MC-LR, MC-RR and MC-YR concentrations already exceeded the new WHO guidelines for microcystin, without the contributions of the other congeners (Appendix A) [39]. However, different geographical origins, species and environmental factors might influence which congeners are more abundant. Analysis of the most common congeners during monitoring is therefore still advisable [20,53,54,55,71,76,77,85,86].

Incorporation of the toxic demethylated congeners in the quantification methodology would be similarly advisable in the future [16].

Besides toxins, *mcyE* was detected in almost 76% of the water samples. Comparing these results is difficult due to differences in the monitoring frequency and amplification method. For instance, the B-BLOOMS2 study found *mcyE* in 89% of their samples, but the number of samples was larger due to more regular monitoring in surface waters prone to bloom events and the addition of occasional bloom samples [23]. Similarly, Moreira et al. (2020) also found variations in *mcyE* presence in various lakes in Portugal. During their sampling from April to September, the *mcyE* detection rate varied from 33.3% to 83.3% for different lakes, but they used a different amplification method than our study [80]. A good correlation between toxin content and the *mcyE* compared to other genes has been shown in the multiple studies reviewed in Pacheco et al. [87]. The authors also pointed out that correlations between *mcyE* copy number and toxin concentration are still controversial. Nonetheless, this data supports the need for our approach of assessing the potential of toxin production by the detection of the *mcyE*.

With the direct Sanger sequencing approach, only the most abundant species would be detected if there was a strong dominance in the sample. Indeed, if the cyanobacteria populations were too heterogeneous, the direct sequencing would fail because the resulting chromatograms would not be interpretable. Therefore, only 45 out of 79 samples produced usable data due to species heterogeneity. Nevertheless, direct Sanger sequencing showed that *Microcystis*, *Dolichospermum*/*Anabaena*, *Synechococcus*/*Cyanobium*, *Planktothrix* and *Aphanizomenon* were generally observed in the samples, as earlier described in the B-BLOOMS2 study. However, *Woronichinia*, also present during B-BLOOMS2, was not detected. Our sampling season, which focused on the warmer months of the year, might have missed the occurrence of *Woronichinia*, which is more prevalent at temperatures below 15 °C [88]. Yet, they might also be present in the more heterogeneous samples but could not be detected with our current methods. Furthermore, the *mcyE* gene was found in samples containing *Cyanobium*, which raises the question of the possible toxicity of this genus, as discussed during B-BLOOMS [23]. Little information is available about the toxicity of picocyanobacteria in eutrophic waters. *Cyanobium* have previously been identified in freshwater ecosystems [89,90], although their toxicity has been put in question. Multiple studies showed low amounts of MCs being produced [15,90,91,92], while the sequenced genomes of *Cyanobium* strains seem to lack a complete NRPS/PKS gene clusters [93].

Amplicon sequencing by a high-throughput sequencing technique is an alternative to direct Sanger sequencing that allows the identification of species in complex populations, allowing the study of heterogeneous samples. However, amplicon sequencing protocol is more expensive and time-consuming. Thus, it was tested for in two water samples, of which one (BL5.29) was used for both sequencing types. This comparison shows that the dominant taxon was quasi-identical with both methods. Indeed, the Sanger sequence was 99.8% similar to the most similar hit for the dominant OTU72 (*A. ellipsoides* Ana HB).

The samples from waterbodies were either collected through one of the regional monitoring programs or following a sporadic bloom notice by the regional environmental agencies in an unmonitored waterbody. The samples from Wallonia and some of the samples from Flanders were obtained from already established monitoring programs for recreational waterbodies, whereas the samples from ponds in Brussels were obtained specifically for the study when a bloom notice was received. In the cases with monitoring programs, the situation is followed by the public authorities and the recreational waterbodies in both regions get regularly closed if (toxic) blooms are observed. In the not monitored ponds in the Brussels region, we found that 66.67% and 16.67% of the samples were above LOQ and 24 µg L^−1^ total microcystin, respectively [67,68]. By comparison, the total MC concentration rose above LOQ and 24 µg L^−1^ total microcystin in 86.08% and 22.78% of the Belgian waterbodies, respectively, at one point in time. Even though the numbers seem to be lower in Brussels compared to the rest of the country, this is artificial. In Brussels, only a small number of waterbodies was sampled. Furthermore, these results were obtained outside the standardized protocol and from only three or fewer time points per site. Therefore, we could have missed bloom peaks during the summer. As we reported, the canal in Brussels also contained a total of 1831.32 µg L^−1^ total microcystin at one sampling spot. Currently, there is a public dynamic in the region to make more waterbodies available for recreational use during the summer. Moreover, the unauthorized use of waterbodies as bathing areas was also common practice during hot summers. Without proper monitoring, this could create a public health risk for humans and domestic animals [9]. In addition, other usages of this water need to be considered. For instance, pond water could be used to irrigate urban vegetable gardens, and water from canals could be used in agriculture. Several research groups have already shown that plants could accumulate MCs when irrigated with contaminated water [45,47,94,95,96]. The connection between water usage and toxin accumulation in plants needs to be further investigated in the future.

Based on our results, it appears that monitoring of any potential bathing sites where unauthorized bathing occurs could be recommended. In the context of Belgium, this would be in the Brussels region. Monitoring could reveal links between public health issues and any potential hazard related to cyanobacterial blooms. The other two Belgian regions monitor their official bathing sites, using different sampling protocols and analysis techniques. Harmonizing the monitoring methods could provide insight into the species and toxin diversity in the blooms in Belgium. Moreover, it might help to uncover environmental drivers that promote the blooms. Techniques used during the monitoring could vary depending on the expertise and resources that are available. Cell counting and species identification are relatively low tech monitoring tools to determine the intensity of the blooms and quantify the potential toxin-producing species, but they are time-consuming and need taxonomic expertise. ELISA and *mcyE* amplification are fast, relatively cheap tests appropriate for screening MCs or MC-production potential, respectively, when a bloom is observed. However, these four techniques need to be supplemented with UHPLC-MS/MS to accurately determine MCs concentrations during and after the bloom to ensure public safety. When toxin equivalency factors become available for the different congeners, UHPLC-MS/MS approaches will also be crucial to accurately determine the risk.

More detailed information about the bloom incidences in a country can also benefit possible mitigation strategies in the future. Preventive strategies could be designed to reduce the influx of nutrients where this is feasible. This strategy requires information about the sources of eutrophication, which is still lacking for most fresh waterbodies in Belgium. During this study; the regional environmental agency listed agriculture, the discharge of purified sewage water, water influx by a canal and the feeding of fish during recreational fishing as potential sources of eutrophication in the Walloon lakes. For smaller ponds in Brussels, an overpopulation of waterfowl can cause nutrient loading due to an abundance of excrement. Another mitigation strategy is hydrogen peroxide treatments. This treatment can eliminate the bloom and MCs but needs to be optimized based on bloom density and the quantity of toxins. For toxin quantification and assistance to prevention, analytical methods, such as the one presented in this paper, would be suitable [97,98,99,100,101,102,103,104,105]. Approaches such as flock lock or flock sink techniques could be a viable solution to prevent bloom incidence by capturing phosphorous on the bottom of larger waterbodies with a greater depth. However, the sediments of shallow recreational lakes might be too frequently disturbed for this approach to be effective [106,107,108,109]. External phosphorous loading in the waterbodies after treatment (e.g., sewage disposal, floods, …) will also undermine the effectiveness of these treatments. To ensure public safety, monitoring of waterbodies will always be necessary and toxin quantification should be included as a part of the monitoring techniques.

## 4. Conclusions

During this study, we validated and used a UHPLC-MS/MS method for the first time to analyze Belgian water samples from the three different regions with an identical method. The microcystin concentrations found clearly illustrate a persistent problem of toxic blooms throughout Belgium with a potential health impact. The three most abundant MCs (MC-RR, MC-LR, MC-YR) contributed the most to the total microcystin concentrations. Our fast and efficient method can be applied to monitoring programs in Belgium and other parts of the world. PCR amplification of the *mcyE* gene linked its occurrence to the toxin presence for 71.43% of the water samples. Moreover, the dominant blooming taxa were also determined in a number of samples. Interestingly, this study also characterized a cyanobacterial bloom in a Belgian canal for the first time. The abundance of water samples that contained MCs shows the need to enlarge the sampling of waterbodies where there could be a risk of human exposure and include them in existing or new monitoring programs.

## 5. Materials and Methods

UPLC/MS grade solvents (Biosolve B.V., Valkenswaard, The Netherlands) were used for extraction or basis for the mobile phase. The MCs standards were ordered as a solid powder from Enzo Life Sciences (Antwerp, Belgium)^®^, except for D-asp-Dhb-MC-RR from Cyano Biotech GmbH (Berlin, Germany) and Dm-MC-RR from Novakits (Nantes, France). They were initially dissolved in 100% methanol and used to prepare mixed stock solutions in 50% methanol with 1% acetic acid. The dissolved cyanotoxin standards were kept at −20 °C. Whatman GF/C grade filters were obtained from Sigma Aldrich (Overijse, Belgium).

### 5.1. Sampling

The sampling was performed from July until mid-September in 2019 at 23 different locations in the three Belgian regions: Wallonia (5 locations), Flanders (7 locations) and Brussels (11 locations). The sampling frequency was dependent on the region, type of waterbodies and access to the lakes (directly or via the environmental agency). Recreational waterbodies are defined as ponds and lakes where bathing is permitted. The water samples were either collected every week or only after a bloom was observed. Each sample was annotated by combining a three digit annotation of the sample site followed by a number giving the week of the year (e.g., XYZ.12). Names for the waterbodies can be found in Appendix A.

In Wallonia, water samples were collected weekly in 5 recreational lakes (I01, I04, E04, B04 and H02), independently of the presence of a bloom following a standardized protocol. The environmental agency Institut Scientifique de Service Public (ISSEP) sampled the surface water with a 500 mL glass bottle at a fixed point. Samples were stored at 4 °C and later transported to Sciensano within three days of the collection for further processing.

In Flanders, water samples were taken from 3 recreational waterbodies (AN1, AN2 and AN3) by the environmental agency Vlaamse Milieu Maatschappij (VMM) only when a bloom was present. The samples were stored in plastic containers at 4 °C before further processing and analysis by Sciensano. Four ponds (GH1, VL1, VL2 and VL3) that were not used for recreation were sampled by Sciensano. GH1 is a sedimentation pond for wastewater, while VL1, VL2 and VL3 are shallow ponds in parks where fishing is allowed. The surface water was sampled with 500 mL sterilized glass bottles. They were processed the same day. In this case, public media had indicated the presence of the blooms. These waterbodies were only sampled a second time by Sciensano if a bloom was still present.

In Brussels, we performed samplings in ponds where bathing is not allowed and thus are not considered as recreational waterbodies. Each waterbody was initially sampled after a bloom notification by the regional environmental agency Environment.Brussels or Port.Brussels. The latter manages the port estate in the Brussels capital region. In total, 8 ponds (BL1-8) and 3 locations in the Brussels canal (BV1-3) were sampled. Each spot was sampled at least a second time independent of bloom presence, except for two spots in the canal. Samples were taken from the surface water with 500 mL sterilized glass bottles. They were processed the same day. An overview of all the sampling sites is shown in Figure 4. An overview of the sample sites with waterbody type can be found Appendix A.

In general, 150 mL of the sample was filtered on a GF/C Whatman^®^ filter under vacuum to collect the biomass. Lower volumes were filtered due to clogging when dealing with high bloom density. The sample filters were weighed before and after filtration to determine the weight of the wet biomass. The sample filters were stored at −20 °C before analysis. Filtration was performed in duplicate. One filter was used for the quantification of the MCs, while the other was used for the molecular work. The filtrates were collected and stored at −20 °C to determine extracellular toxin concentration.

### 5.2. Quantitative Analysis of Microcystin Congeners

#### 5.2.1. Intracellular and Extracellular Microcystin Extraction

Only the most common toxins in Europe (MCs) were selected for our quantification method. Earlier studies in Belgium suggested that these toxins are the most prevalent public health threat [20,23]. To properly validate the method, only commercially available MCs were selected.

The method used for analysis was validated in house. Results of the validation can be found in Appendix A. The filters, containing biomass, used for toxin extraction underwent a freeze-thaw step and liquid extraction. When the filters were initially stored at −20 °C, they only need to be defrosted. The filters were cut in half and weighted. For the liquid extraction, 4.5 mL 80% methanol was added together with the filter in 50 mL plastic tubes. Solvent and biomass contact was increased by regularly mixing during 1 h. The samples were centrifuged for 10 min at 3900 rpm.

The extract was filtered through a Phenomenex 0.2 µm RC syringe filter (Utrecht, The Netherlands) to remove debris. Samples were stored in a 15 mL plastic tube at −20 °C. Samples with high concentrations of the MCs were diluted after the initial analysis to fit within the range of the calibration curve. The calibration curve was made in a blank matrix.

The sample filtrates (extracellular fraction) were also purified using a Phenomenex 0.2 µm RC syringe filter and analyzed separately through direct injection of 10 µL the Xevo TQ-S, similar to Turner et al., 2018 [51].

#### 5.2.2. Detection and Quantification of Cyanotoxins

The detection and quantification parameters were identical for intra- and extracellular toxins analysis. A Waters Acquity UPLC H-class (Eten-Leur, The Netherlands) connected to a Waters XEVO TQ-S was used for the detection of the cyanotoxins. A 1.7 µm, 2.1 mm × 100 mm Waters Acquity BEH C18 column fitted with a Waters Acquity BEH C18 1.7 µM VANGUARD PRE-Col separated the toxins under the influence of a gradient elution program. The fraction of acetonitrile (B) in the eluent changed as followed: 0 min, 2% B; 1.00 min, 40% B; 7.00 min, 55% B; 7.20 min, 98% B; 8.00 min, 98% B; 9.00 min; 2% B; 12 min, 2% B. Both Organic and water phases were supplemented with 0.025% formic acid. The flow rate was 0.5 mL min^−1^. The column was heated to 60 °C, and 10 µL of sample was injected.

Multiple reaction monitoring (MRM) was then used to detect the toxins by selectively quantifying compounds within complex samples. The triple quadrupole MS initially targeted the ions corresponding to the toxins of interest, referred to as the “precursor ion”. Two product ions from the collision induced fragmentation were selected. One was used for quantification of the cyanotoxin, the other as a qualifier. The MS parameters were set according to the literature data and optimized to the instrument setting (Table 2).

After quantification, the concentration for each cyanotoxin was recalculated to µg L^−1^, corrected with the mass of the original mass of the filter and the filtered volume of the sample. The concentration of each MC in the filtrate (extracellular fraction) was added to the final concentration of the MC extracted from the biomass. Thereafter, the sum of all the congeners was calculated to provide a µg L^−1^ total microcystin value.

Congener proportions to the total MCs concentration were calculated in each sample. The differences in proportions for the separate congener were then compared for Belgium using the Wilcoxon test at α < 0.05. Additionally, the same statistical test was used to compare the difference in congener proportions for samples containing MCs concentration higher and lower than 24 µg L^−1^ total microcystin, separately. The samples without MCs were excluded. Proportions of MC-LR, MC-RR and MC-YR were also compared between the two concentration ranges.

### 5.3. Molecular Analysis of the 16S rRNA and the mcyE Gene

#### 5.3.1. DNA Extraction

First, 0.8 mL lysis buffer (40 mM EDTA 5, 50nM Tris-HCl, 0.75 M sucrose) was added to each sample filter (containing biomass), and a bead-beating step (at 30 m s^−1^ for 30 s) was performed. Then, a lysozyme (Sigma-Aldrich, St. Louis, MI, USA) (20 mg mL^−1^) digestion for 30 min at 37 °C was followed by a treatment with 22.22 mg mL^−1^ proteinase K (Macherey-Nagel, Düren, Germany), supplemented with 80 µL SDS (100%), for 2 h at 55 °C. The lysate was transferred to a new Eppendorf tube. Subsequently, the filters were rinsed with 1 mL lysis buffer during a 10 min incubation at 55 °C. The second lysate was stored in another Eppendorf tube.

A Phenol-chloroform-isoamyl alcohol solution (25:24:1, pH 8) (VWR, Leuven, Belgium) was added in an equal volume to the extract volume (V:V) to both lysates. Next, the samples were centrifuged at 14,000× *g* for 15 min. The upper phase of each tube was transferred to a new Eppendorf tube, and chloroform-isoamyl alcohol (24:1, pH 8) was added V:V. The tubes were centrifuged again at 14,000× *g* for 15 min to collect the upper phase. For each sample, the two lysates were combined

Finally, the DNA was precipitated with 0.1 V:V of sodium acetate (3 M, pH 5.2) and 0.6 V:V of cold isopropanol. After centrifugation, the DNA was rinsed once with 300 µL ice-cold ethanol (Merck, Branchburg, NJ, USA) (100%) and once with 300 µL ice-cold ethanol (70%). The supernatant was removed, and the pellet was air dried. Finally, the DNA pellet was dissolved in 100 µL TE buffer (10mM Tris-HCl and 1mM EDTA, pH 8) and stored at −20 °C.

#### 5.3.2. Gene Amplification of Partial rRNA and mcyE Gene Sequences

For the rRNA gene sequences, two protocols were tested. A long rRNA fragment was amplified with the cyano-specific primers 359F/23S30R [110] using the SuperTaq Plus^©^ enzyme (HT Biotechnology, Cambridge, UK), buffer and dNTPs obtained from SpharoQ (NL). The amplification program was 95 °C—5 min, 35 times; 95 °C—30 s, 57 °C—45 s, 68 °C—1 min; followed by 69 °C—5 min, 16 °C—infinite. As a shorter PCR product could give a higher amplification efficiency, the primer pair 359F/781R [111] was tested later. However, the SuperTaq Plus^©^ enzyme was no longer commercialized and was replaced by the Q5 High Fidelity polymerase (New England Biolabs, Ipswich, MA, USA) for the majority of the PCR reactions. The amplification program was: 98 °C—5 min, 35 times; 98 °C—30 s, 65 °C—45 s, 72 °C—1 min; 72 °C—5 min, 16 °C—infinite. The *mcyE* gene involved in the production of MCs was amplified with the primer pair mcyEF2/mcyER4 [112] using the SuperTaq Plus© enzyme. The amplification program was 94 °C—3 min, 30 times; 94 °C—30 s, 57 °C—45 s, 68 °C—1 min, followed by 68 °C—10 min and 16 °C infinite, as described in the final report of B-BLOOMS2 [23]. Amplifications were performed in a Thermal cycler T100 (Bio-Rad, Hercules, CA, USA). The presence of PCR products of the right size was visualized by electrophoresis on a 1.5% agarose gel during a 95 min run at 90 V.

#### 5.3.3. Sanger Sequencing and Sequence Analysis

After PCR, the 16S rRNA amplicons were sent for Sanger sequencing with primers 359F, 781R or 23S30R at Giga Genomics (ULiege) [110,111]. Some sequences were of bad quality, which prohibited further analysis. These sequences probably resulted from a mixture of organisms without clear dominance by one taxon. The forward and reverse sequences were not obtained in all cases for each PCR product, and therefore, the individual sequences of a single strand were used for further analysis, admitting that some sequencing errors might be present but that the quality would be sufficient to determine the dominant genus. In three cases, the sequences obtained on different PCR products (short or longer ones) for the same sample were affiliated to different genera. The sequences used during the further analysis can be found in the Appendix A.

The NCBI nucleotide BLAST (basic local alignment search tool) was used to identify the most closely related strain sequences for the 16S rRNA sequences, using individual sequences obtained by the different primers tested and the identification was based on this data, as shown in Appendix A.

#### 5.3.4. Amplicon Sequencing with the Illumina Technique

For samples BL5.29 and VL1.36, partial 16S rRNA gene sequences were obtained by PCR using the primer set CYA359F and CYA781Ra/CYA781Rb, which amplifies the V3-V4 region of the cyanobacterial 16S rRNA gene [111]. Primers were modified to include a 10-bp sample-specific barcode tag at the 5′ end to allow samples to be multiplexed for sequencing. PCR reactions were performed in triplicates in order to minimize the influence of amplification biases. These were pooled to equivalent concentrations and purified using the NucleoSpin^®^ Gel and PCR Clean-up kit (Macherey-Nagel, Düren, Germany). Purified samples were sent to Genewiz (South Plainfield, NJ, USA), where sequencing adapters were ligated to the amplicons and sequencing was performed using Illumina MiSeq (Illumina, San Diego, CA, USA) using 2 × 300 bp paired-end libraries. The bioinformatic analysis is adapted from a validated method by Pessi et al. and consists of processing raw reads to remove chimeric sequences, followed by the clustering into an operational taxonomic unit (OTU) [113]. Briefly, paired-end reads were merged, filtered and only reads containing both barcodes in the 3′ and 5′ ends were kept. Two and zero mismatches were allowed to the primer and barcode sequences, respectively, and reads with a maximum expected error of more than 0,5 and a length of less than 370 bp were removed. Singletons were removed, and remaining quality-filtered sequences were denoised to remove chimaeras and sequencing errors using unoise3 [114]. The denoised operational taxonomic units (ZOTUs) obtained were then clustered into OTUs at a 99% similarity threshold [115]. The representative sequence of an OTU is the most abundant unique sequence of each OTU cluster. Taxonomic classification was performed by extracting from Genbank the most closely related sequences of each OTU using BLAST.

## Figures and Tables

**Figure 1 toxins-14-00061-f001:**
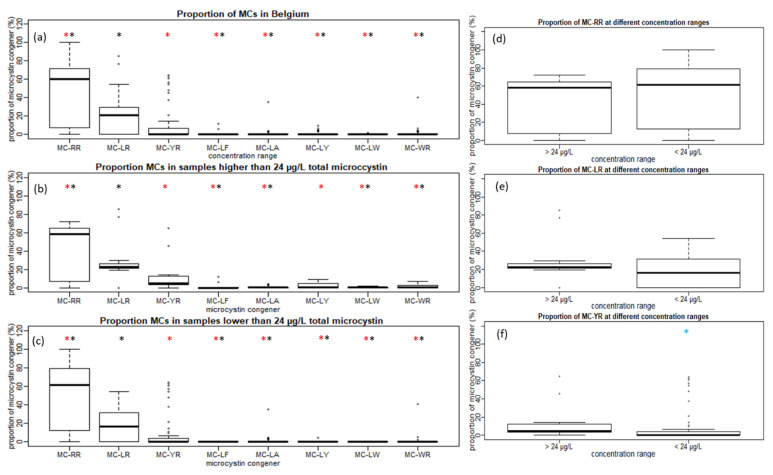
(**a**) The distribution of the proportion of microcystin congeners (MCs) calculated at an individual sample level for all Belgian samples. (**b**) Samples with concentrations higher than 24 µg L^−1^ total microcystin. (**c**) Samples with concentrations lower than 24 µg L^−1^ total microcystin. (**d**) Proportions of MC-RR are compared in samples below and above the World Health Organization (WHO) guideline value. (**e**) Proportions of MC-LR are compared in samples below and above the WHO guideline value. (**f**) Proportions of MC-YR are compared in samples below and above the WHO guideline value. * Proportion of MC is significantly different from MC-LR at α < 0.05 using the Wilcoxon test. * Proportion of MC is significantly different from MC-YR at α < 0.05 using the Wilcoxon test. * Proportion of MC is significantly different from the proportion of MC at concentration range > 24 µg L^−1^ total microcystin with α < 0.05 using the Wilcoxon test.

**Figure 2 toxins-14-00061-f002:**
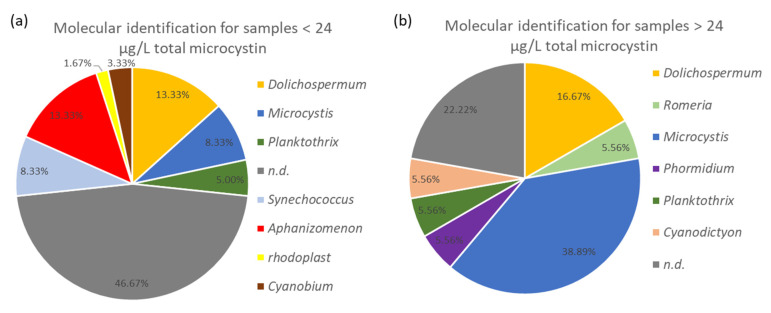
Identification of species using direct Sanger sequencing and BLAST analysis. Samples were divided based on the WHO guideline value for recreational ponds (24 µg L^−1^ total microcystin (MC)). The “n.d.” abbreviation refers to not exploitable 16S rRNA sequences. (**a**) Species distribution for samples containing total MCs concentration below the WHO guideline value. (**b**) Species distribution for samples containing total MCs concentration above the WHO guideline value.

**Figure 3 toxins-14-00061-f003:**
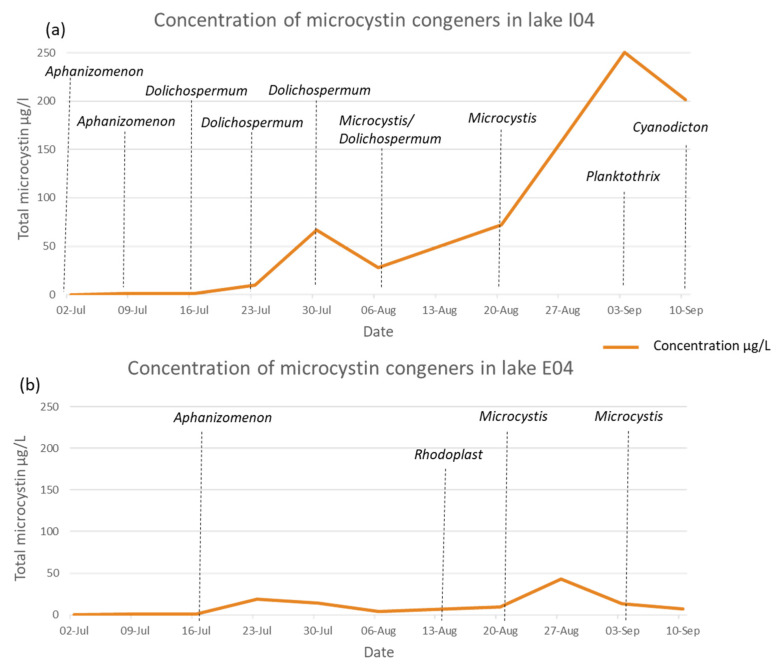
(**a**) Evolution of total microcystin concentrations in Lake I04 (lac de Bambois, Fosses-La-Ville) during the summer of 2019. Dominant genera detected in the samples are also indicated. (**b**) Evolution of total microcystin concentrations in Lake E04 (Grand large, Mons) during the summer of 2019. Dominant genera detected in the samples are also indicated.

**Figure 4 toxins-14-00061-f004:**
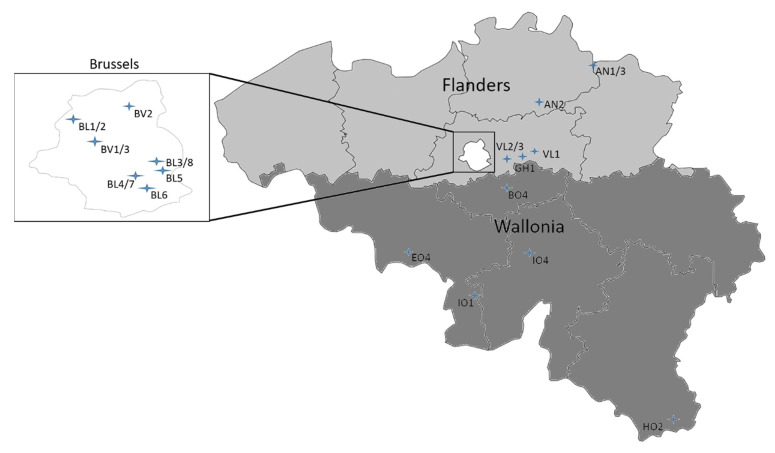
Map of Belgium showing the sample sites. The first three letters of the sample names are used as abbreviation. In Flanders, 7 sites were sampled (AN1-3, VL1-3 and GH1). In Wallonia, 5 recreational lakes were sampled (I01, I04, E04, B04 and H02). For clarity, the Brussels region is enlarged. Here 8 ponds were sampled (BL1-8), as well as the Brussels canal at 3 different sites (BV1-3). Place names for the waterbodies and their type can be found in Appendix A.

**Table 1 toxins-14-00061-t001:** Overview of precursor ion, product ions and limit of detection for not validated microcystin congeners. Additionally, the table also includes the detection frequency of the congeners in the analyzed samples at different total microcystin concentrations.

Toxins	MC-HtyR	dm MC-LR/D-asp MC-LR	D-asp-Dhb MC-RR/dm MC-RR	MC-HilR
Precursor ion	1059.5	981.14	512.7	505.3
Product ions	106.9; 135.27	106.8; 135.07	103.2; 135.13	126.99; 134.92
Limit of Detection (µg L^−1^)	0.1	0.1	0.1	0.1
All samples	13.92%	53.16%	77.22%	34.18%
Samples < 1 µg L^−1^ total microcystin	0.00%	9.38%	50.00%	0.00%
Samples > 1 µg L^−1^ total microcystin	23.40%	82.98%	95.74%	57.45%
Samples < 24 µg L^−1^ total microcystin	25.00%	100.00%	100.00%	100.00%

**Table 2 toxins-14-00061-t002:** MS/MS parameters for eight microcystin congeners (MCs).

Toxins	Precursor Ion m/z	Quantifier Ion m/z	Collision Energy (eV)	Cone Voltage (V)	Qualifier Ion m/z	Collison Energy (eV)	Cone Voltage (V)
MC-LR	995.4	135.0	70	80	213.1	60	80
MC-RR	519.8	134.8	30	50	107.2	60	50
MC-YR	1045.5	135.3	80	60	212.9	60	60
MC-WR	1068.4	135.3	70	100	213.1	60	100
MC-LY	1002.4	135.4	60	50	213.0	50	50
MC-LA	910.3	135.1	60	50	107.1	80	50
MC-LF	986.3	135.0	60	70	213.1	60	70
MC-LW	1025.4	134.9	60	60	213.1	50	60

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
