# Peer review of "A Summer of Cyanobacterial Blooms in Belgian Waterbodies: Microcystin Quantification and Molecular Characterizations"

_toxins, 2022, doi:10.3390/toxins14010061_

Round 1

Reviewer 1 Report

Manuscript: A summer of cyanobacterial blooms in Belgian waterbodies: analytical and molecular characterizations

Comment about the title: I don't think "analytical" is a suitable term to separate toxins analysis from molecular ones. Maybe you should use "chemical" or "cyanotoxins detection".

General comments: The manuscript highlights the occurrence of MCs congeners and a molecular survey of the main bloom-forming toxic cyanobacteria in several waterbodies along Belgium. The main goal of the study was to identify waterbodies which are (or not) in line with WHO guidelines regarding MCs limits for recreational activities. Also, it important to mention that the study was conducted with modern techniques and a high analytical accuracy. It is relevant study that rise the discussion on advances of cyanobacterial monitoring in waters for supply and recreational using, especially to policy makers, but some writing issues address some inconsistency which must be carefully revised.

Please, find bellow the specific comments with more information.

Introduction

Lines 34 – 36: Please, provide some information regarding the impacts on the water quality in these water bodies. Is there any sewage disposal with further eutrophication signal? It is welcome to introduce cyanobacterial blooms.

Lines 43 – 44: "…morphological diversity of cyanobacterial blooms" seems to have no sense. Please, re-write to make it clearer.

Lines 44 – 54: Here we can note that this study have been conducted in Belgium since 90's. Therefore, what kind of advance the present study brings to the knowledge of Belgian cyanoHABs? What is the gap?

Lines 64 – 65: Please, first describe the action mechanism of MCs. Also, it is important to consider that, besides primarily affecting the liver, these toxins can also affects other organs such as lungs and kidneys.

Another suggestion is to provide some information regarding exposure (e.g., recreational, dietary, occupational), since most of the introduction points out the WHO guidelines values and exposure human risk.

Lines 68 – 69: This is a very good point. Maybe you should provide further information on the chemical diversity of MCs congeners and toxicity variation (e.g., hydrophobicity, pKa).

Line 89: According to the exposed, the characterization has already been done. "to monitor" makes more sense.

Results

Line 103: In my opinion, describing first the cyanobacterial species and their relative abundance identified along the water samples make more sense, before showing toxins data.  

Line 106: Is this unit (µg kg-1) correct?

Lines 138 – 140: It would be welcome to evaluate if at this same point a higher cyanobacterial diversity (or co-dominant taxa) has occurred. Do these toxins were produced by a single dominant species or more than one?

Lines 208 – 212: Did you mean "dominant" when mentioning "containing" with regards to the cyanobacterial taxa?

Discussion

General comment

Firstly, make clear that cyanoHABs and high levels of MCs were remarkable in the summer. The title of the manuscript points this out.

Also, a general comment is that the discussion must be improved for the manuscript not to look like a technical report, although much the information present here was inspired by a pre-existing monitoring program, frequently mentioned in the text.

Lines 272 – 273: Please, provide further information on why higher MC content was observed in Flanders. Is this a consequence of more dense blooms? Is there any phytoplankton biomass data?

Lines 297 – 303: This statement is not fully supported by the results. It is important to discuss that the qualitative change in MCs present in cyanoHABs follows primarily the dynamics/diversity of MC-producing strains/species.

Furthermore, the total cyanobacterial biomass in these lakes must be considered and discussed, besides some considerations regarding cell toxicity (e.g., MC cell content; high toxin concentration under a low biomass suggesting cells with a higher toxin production). 

See the rational discussed in Orr et al. (2018) Journal of Oceanology & Limnology.

Lines 310 – 311: Would the water solubility of these MCs congeners explain their abundance, since most of those frequently reported for waters are hydrophilic?

Would be the others (MC-LA, -LW, -LF...) associated to biological matrices?

Line 314: Remove "in the future". The sentence is redundant.

Lines 315 – 322: Please, provide further information on mcyE and its reliability as a MC-producing cyanobacterial marker over other MC genes.

I recommend you read Pacheco et al. (2016) Toxins.

Line 332: Write the species name in italic.

Lines 334 – 336: This is something poorly discussed in the literature in general and deserves an attention here. Is picoplanktonic cyanobacteria an important source of cyanotoxins in eutrophic waters?

Lines 374 – 379: Cell counting is also fundamental. Please, do not forget to mention it.

Line 386: Replace "approaches" by "techniques".

Lines 388 – 389: Also, it is important to consider the if the P source is allochthonous or autochthonous, since flock-lock or sink methods focuses more in controlling the internal load of phosphorus.

Lines 389 – 340: This sentence is a bit confuse. Please clarify.

Conclusions

Please, mention that the results identified the need of monitoring other waterbodies in Belgium which can be potentially used for recreational activities and represent a human risk of cyanotoxins exposure.

Line 393: Remove "for the first time".

Line 399: Replace “presence” by "occurrence".

Materials and Methods

Line 457: Why did you focus only on MCs? Make it clear in the methods.

Line 458: This section should also describe the MCs recovering from dissolved fraction.

Line 488 – 489: Again… How did you recover the MCs from the dissolved fraction? Please, describe it in the methods section.

Line 551: Does bioinformatics analysis is also included in this section?

Reviewer 2 Report

The manuscript is very well written and contains publishable data. However, small modifications are necessary before publication (see attached document). 

Reviewer 3 Report

The manuscript "A summer of cyanobacterial blooms in Belgian waterbodies: analytical and molecular characterizations" has studied the presence and concentrations of the most frequent microcystins congeners in different water samples of Belgium, as well as the dominant species in these samples. 

The article is well elaborated and provides results of interest that be worthy of  its publication.

However, minor corrections and revisions are necessary.

  • Page 2, line 76. Add WHO 2020 reference.
  • Page 2, line 86. The authors said: “the mean body weight of children between 6 and 10 kg …”. Did they mean, 6 and 10 kg or 6 to 10 years?.

Reviewer 4 Report

The article entitled "A summer of cyanobacterial blooms in Belgian waterbodies: analytical and molecular characterizations" deals with a contemporary topic. The problem with increased cyanobacterial blooms and cyanotoxins worldwide is a fact that could be partly explained by global climate change. The risk for the human health and the consequences for the affected water bodies point the attention of researchers to study the causes of blooming, cyanotoxins released from the blooms, the monitoring that is required and ways to solve the problem. Given the above, I find the article useful. However, I do not think that it represents something new for the science. The authors use generally accepted methods for analysis, and the data, though much, are local.

I have the following questions and remarks to the authors:

  • In which year(s) were the samples collected? This is not clear. Please, add this information in the section Materials and methods.
  • Are physicochemical parameters (at least temperature) measured during the sampling? Are there any correlations with the blooming, the presence of cyanotoxins and their amount? If there are, it is good to be included.
  • It is not clear whether there were blooms during the sampling, and if so, how intense were these blooms. Moreover, the title of the article emphasizes the blooms.
  • The dominant cyanobacterial species have been determined by 16S RNA, but has their presence in the sample been confirmed by microscopic analysis? If yes, it is good to present pictures for the morphology.

• How species were defined as dominant? Based on which criteria?

Round 2

Reviewer 1 Report

Comment about the title: Please, replace “microcystin” by “microcystins”.

General comments: The authors have made most of the suggested modifications to improve the manuscript, but some little writing issues are still to be revised. Also, somehow, authors’ responses identified by the line numbers were not in line with those in the paper, which have difficulted the revision.   

Please, find bellow some specific comments:

Introduction:

Lines 39 – 41: The sentence has writing issues. Please, rewrite.

Lines 67 – 68: “…study by providing new toxin, molecular and occurrence data…” needs complement. “Occurrence” of what?

Lines 81 – 83: These sentences can still be improved. Please, write this rational in one sentence and provide some information on toxicity variation regarding congeners chemical diversity. In my opinion this information is not out of the focus of the paper, since the authors have analyzed a range of MCs and associated these findings with human exposure risk upon recreational activity.

Results

Lines 119 – 123: In my opinion this topic is unnecessary to describe cyanotoxins quali-quantitative analysis. The results may initiate as “2.1 Toxin quantification”. Chemical analysis of water samples reminds more to physical-chemical parameters, which would be very welcome (at least temperature and nutrients) to be shown in the results.

Lines 159 – 161: Still, in my opinion this result still has some inconsistency if not related to cyanobacterial diversity in the sample. It is important to argue if a higher MCs diversity is associated to different cyanobacterial taxa or, e.g., only Microcystis. At this point, the authors’ response letter brings some information about sequencing data that could be added to the results to clarify this issue.   

Lines 292 – 296: The previous question on the higher MCs content found in Flanders is still to be clarified in the text. Is this a consequence of more dense blooms? If no cell counting data is available, you can at least use fresh biomass data obtained from filters weight.

Reviewer 2 Report

No more comments

Reviewer 4 Report

As already mentioned, the data do not represent anything new or have any scientific contribution. The methodology used to determine the dominant species does not comply with generally accepted standards. Only the presence of one group of cyanotoxins without proven correlation with cyanobacterial blooms or environmental factors was presented. Moreover, this is a local study and these toxins are not reported for the first time.

Round 3

Reviewer 1 Report

The authors have made most of the required modifications. I only would like to suggest them to check carefully the fresh weight values of filtered biomass because they seem low regarding a 1-L bloom sample.

Furthermore, in my opinion the paper can be accepted in the present form. 

This manuscript is a resubmission of an earlier submission. The following is a list of the peer review reports and author responses from that submission.

Round 1

Reviewer 1 Report

  1. Throughout the text: It is totally misleading to call the sum of concentrations of microcystin congeners "MC-LR Equiv" when the results have been generated by UPLC-MS/MS. "MC-LR Equiv" would only be meaningful in toxicological analysis (protein phosphatase inhibition assay) when MC-LR is used as standard or in ELISA where antibodies have been generated against a MC-LR conjugate. No agency is encouraging the use of the term "MC-LR Equiv" in the way chosen by the authors. Total microcystin is a simple and descriptive term.
  2. It is a serious omission not to include demethylated MC-RR, -LR and -YR among the congeners analysed by UPLC-MS/MS. In particular Planktothrix produces of demethyl-MC-RR as the main/sole toxin. Are there MS full scans available where you could look for the demethyl variants especially in the Planktothrix-dominated samples?

Reviewer 2 Report

This study present the monitoring of cyanobacterial blooms in Flanders and Wallonia. This study indicates that Belgian waterbodies are poorly monitored for cyanotoxins and toxic cyanobacteria. Improving the monitoring of toxic cyanobacteria and their toxin is important especially in a context where in some countries there is limited monitoring and episodes of harmful blooms. Unfortunately, this paper lacks information regarding the validation of analytical methods to support the results, lacks novelty and should be revised before reconsidering its publication.

  1. Please specify the novelty of this study at the end of the introduction. It is not clear in the statement of the aim of the study how the results obtained will be innovative for monitoring cyanotoxins in Belgium (analytical methods, cyanotoxins studied, cyanobacteria studied, number of sites, spatio-temporal studies?).
  2. Line 129: 628.73 µg L-1, please put -1 in superscript.
  3. Line 151: The same results, […], were obtained, please modify.
  4. Figure 3: For a clearer presentation of the results, it would be better to put the dates on the x axis instead of weeks. Error bars should also be added to the results to show variability, and this variability should be showed in the results too.
  5. Line 238: The fact that LC-MSMS was firstly used for the monitoring of cyanotoxins in Belgium should be better reflected in the manuscript since this method is already widely used and preferred in monitoring programs including the US EPA. This improvement is important to put forward.
  6. The microcystins concentrations are very high and abundant. It would be interesting to compare your results with the concentrations found in studies and in other countries to better contextualize the health hazard of such concentrations (up to 600 µgL-1). As for example, if these waters are used as drinking water sources, are the water treatments sufficient?
  7. A conclusion section is missing and the interest and novelty of this study should be put forward to improve monitoring of toxic cyanobacteria and their toxins.
  8. Appendix A is quite heavy in the main manuscript, it should be added in the Supplementary materials with tables in landscape mode.
  9. Line 421: Did you sampled duplicates for your study or use one sample and filtered twice? If only one sample was taken during the sampling, method validation should be provided to support the validity of the results.
  10. It is unclear if the analytical methods were developed for this study or validated methods were used. The method validation values must be accessible to ensure the validity of the results, one must either add references to the methods in the material and method section or provide the validation results.

Reviewer 3 Report

Dear Authors,

I carefully read the MS « A summer of cyanobacterial blooms in belgian waterbodies : analytical and molecular characterizations » based on the comparison of 5 lakes and 78 samples in MC contents, dominant cyanobacterial genera and some simple molecular analyses of mcyE gene. Even if a valuable effort is made in this country, the major issue came finally from the local interest of this investigation and the use of some limiting and controversial methods carrying out to analyse the possible presence of microcystins by a simple molecular detection of one gene implied in the MC synsthesis (mcyE). While it is now known as a limiting and inappropriate critera, as the molecular detection of this gene is not correlated to the effective production and quantitatively presence of MC in waters. Thus this method alone is inaccurate to evaluate MC content in waterbodies. For the taxonomical investigation, they provided some Sanger results which are not so performant to analyse the cyanobactreial community and dominance as the authors mentioned themselves in the discussion. For the analytical method (MS/MS) the authors detected only the standards MC congeners without the demethylated ones, which can underestimated significantly the total MCs concentration in water samples (especially when Planktothrix is dominant in water).

Consequently, the two major issues concerned 1) the methodologies used for the investigation ie limited, inaccurate and not the best fit to the general purpose, and 2) the rather local interest for the professional agencies in Belgium, which is not really adequate to the topics of this high level journal.

Some minor issues can be further noted throughout the MS :

- The historical introduction is too long and not so interesting for the large audience

- You mentioned the analyse of nodularin, but no result of this toxin, which could not be found in belgium freshwaters, could it…

- Fig 1 : Are the distinction between the MCs up to 24µg and < this value with the total MC so important to mention them in one figure ?

- Fig 2 : the « no result » means that the Sanger sequencing failed ? Or not exploitable ?

- Fig 3 : the different weeks don’t mean anything and should be replaced by the season or month to facilitate the reading. Fig 3 b : why is there so few dominant cyanobacteria detected in each week ?

Please keep the same legend for the y axis between a and b to better compare MC values.

- The « results » section is not so informative and can be more detailed between lakes, samples, seasons, with cyanobacterial dominance and the possible correlation between the total MC contents…

- Discussion : remove the NOD detail, as you did not provided any results on it. This section could be more focused on your results and more concise as well.